# Antioxidant and Anti-Inflammatory Activity of Cell Suspension Culture Extracts of *Plukenetia carabiasiae*

**DOI:** 10.3390/ijms262412190

**Published:** 2025-12-18

**Authors:** Janet María León-Morales, Soledad García-Morales, Maura Téllez-Téllez, Alexandre Cardoso-Taketa, Mónica Morales-Aguilar, Celso Gutiérrez-Báez, Anabel Ortiz-Caltempa

**Affiliations:** 1Coordinación Académica Región Altiplano Oeste, Universidad Autónoma de San Luis Potosí, Salinas de Hidalgo 78600, San Luis Potosí, Mexico; janet.leon@uaslp.mx; 2SECIHTI-Centro de Investigación y Asistencia en Tecnología y Diseño del Estado de Jalisco, A.C., Zapopan 45019, Jalisco, Mexico; smorales@ciatej.mx; 3Centro de Investigaciones Biológicas, Universidad Autónoma del Estado de Morelos, Cuernavaca 62209, Morelos, Mexico; maura.tellez@uaem.mx; 4Centro de Investigación en Biotecnología, Universidad Autónoma del Estado de Morelos, Cuernavaca 62209, Morelos, Mexico; ataketa@uaem.mx (A.C.-T.); moraaguifcb@gmail.com (M.M.-A.); 5Herbario de la Universidad Autónoma de Campeche, Centro de Investigaciones Históricas y Sociales, Universidad Autónoma de Campeche, Campeche 24030, Campeche, Mexico; cgutierr@uacam.mx

**Keywords:** micropropagation, phenolic compounds, antiradical activity, nitric oxide, secondary metabolites

## Abstract

The genus *Plukenetia* includes lianas or vines with oleaginous seeds rich in omega-3 and omega-6 fatty acids, proteins, and vitamin E, and the presence of flavonoids, steroids, and terpenoids has also been reported in leaves. Several species of *Plukenetia* have traditionally been cultivated in their native distribution areas, and their propagation is usually by seed. The aim of this work was to establish callus and cell suspension cultures of *P. carabiaseae*, an endemic species of Mexico, for the evaluation of the in vitro antioxidant and anti-inflammatory potential of its extracts. Three light conditions were evaluated for the establishment of *P. carabiaseae* callus lines from leaf explants. Friable calluses obtained under constant light were used to initiate a cell suspension cultures in Gamborg basal (B5) medium supplemented with 2,4-dichlorophenoxyacetic acid (2,4-D) and kinetin (CIN), as growth regulators. After 35 days of cultivation, different polarity extracts from biomass were obtained, showing that the acetone extract had the highest antioxidant activity and a high total phenolic content (30.57 mg of gallic acid equivalent (GAE)/g dry weight). The anti-inflammatory activity of the methanolic extract, evaluated in murine macrophages induced with bacterial lipopolysaccharides, was dose-dependent, without cytotoxic effects. This is the first report of the establishment of *P. carabiasiae* cell suspension culture and demonstrates its potential as a biotechnological source of antioxidant and anti-inflammatory metabolites.

## 1. Introduction

The genus *Plukenetia* (Euphorbiaceae) comprises 25 species of lianas or vines distributed in the tropical regions of America, Africa, Asia, and Madagascar [1]. *P. volubilis* L., known as *Sacha inchi*, is the species most studied from a therapeutic and nutritional point of view. Some activities reported for this species include antioxidant [2], antiproliferative [3], and anti-inflammatory [4]. In Mexico, the presence of three species, *P. penninervia*, *P. stipellata*, and *P. carabiasiae*, has been reported [5]. The latter is the only species endemic to Mexico with a limited geographical scope and classified as an endangered species and critically endangered, according to the calculation of area of occupancy (AOO) and extent of occurrence (EOO), two important criteria of the International Union for Conservation of Nature to classify species by their potential risk of extinction [6]. The studies reported to *P. carabiasiae* are limited to the morphological description and distribution [1,5,7]. Besides, the species of the genus are known to produce seeds of different sizes that are rich in polyunsaturated fatty acids, such as omega-3 (128–160 g/kg) and -6 (124–141 g/kg) as well as by their protein content (290 g/kg). β-sitosterol is a major phytosterol in the ethanolic extract of leaves and roots of *P. volubilis*, as well as in seed oil (45.2–53.2 mg/100 g seed), highlighting the abundance of linolenic acid and β-carotene in the latter [8,9]. Traditionally, these oilseeds are consumed roasted, or their oil is used in the preparation of meals in the Amazon region, and dried leaves are used to make tea in Thailand [10].

Research on species of the genus *Plukenetia* has focused on the characterization and use of seeds oil. Furthermore, the presence of other bioactive compounds in the seed of *P. volubilis* has also been reported, such as total phenols (64.6–80.0 mg GAE/100 g of seeds) and tocopherols (79–137 mg/100 g of seeds). In the latter, γ- and δ-tocopherols were found to be the major compounds [9]. The content of these phytochemicals has been linked to the antioxidant capacity in the seed. Also, other parts of the plant, such as fruit and leaves, have compounds with significant pharmacological properties. In a preliminary screening of *P. volubilis* the presence of flavonoids, steroids, and terpenoids in leaf extracts of different polarities were reported, with the most polar extracts presenting the best antioxidant activity [3]. Some flavonoids derived from kaempferol were identified and astragalin isolated from the leaf infusion with anti-*Helicobacter pylori* activity [11]. Fresh leaves of *P. volubilis* are also rich in saponins (301 mg/kg DW) and alkaloids (146 mg/kg DW), and these contents are reduced after the heating process [12]. The anti-arthritis activity exhibited by the ethanolic leaf extract in the murine model was carried out through the modulation of inflammatory cytokines [13]. Considering the above, the presence of bioactive metabolites in the genus *Plukenetia* have interesting pharmacological potential still unexplored.

The species of this genus have been grown from seed, and the current propagation has been carried out by cuttings to reduce the limitations imposed by the low germination rate, long life cycles, or irregular fruiting and flowering, in addition to maintaining the parental genotype. A widely used technology for the stable production of bioactive compounds from plants at any time of the year, as well as the implementation of strategies to enhance their production, using elicitation and scale-up in bioreactors, is the in vitro culture, which allows the propagation of seedlings or plant tissues under controlled conditions and confined spaces, in addition to the rational use of biodiversity [14]. Some reports on the species *P. volubilis* include in vitro germination of seeds to obtain seedlings [15], micropropagation from axillary buds [16], and induction of calluses from leaf and ovule explants [17]. Due to the lack of pharmacological studies of *P. carabiasiae* and the scarcity of wild specimens, the objective of this study was to evaluate the antioxidant and anti-inflammatory potential of the *P. carabiasiae* cell suspension culture extracts established from leaf explants.

## 2. Results

### 2.1. Establishment of Callus Culture

Figure 1a,b showed the tetralobed fruits collected from *P. carabiasiae*, with four seeds. The seeds were scarified to remove the coat (Figure 1c) for subsequent germination.

Three lines of *P. carabiasiae* calluses were established and grown under constant light, photoperiod, and in darkness. The most friable calluses were observed in the line maintained in constant light and with yellow pigmentation (Figure 2a), followed by calluses grown in darkness (Figure 2c), highlighting their low pigmentation.

The three callus lines showed accelerated growth until reaching the maximum biomass yield on day 33, with values of 7.7, 6.9, and 5.6 g dry weight (DW) for calluses grown in constant light, darkness, and photoperiod, respectively (Figure 3). Subsequently, growth was maintained until the 42 days of cultivation.

### 2.2. Growth Kinetics of Cell Suspension Culture

The cell suspension culture of *P. carabiasiae* was established from the line of friable calluses grown under constant light (Figure 2a). Callus inoculation was performed in flasks with side baffles containing the same Gamborg basal (B5) medium added with the 2,4-D: CIN growth regulators in which the calluses grew, but without the addition of phytagel (Figure 4a). When complete callus disaggregation was observed, the established suspension culture was transferred to Erlenmeyer flasks (Figure 4b).

In the growth kinetics, fast growth was observed with an exponential phase from day 0 to day 12 of culture, without an adaptation stage. The maximum biomass yield was reached on day 12 with a value of 16.07 g/L DW, which represented 4.3 times more than the initial inoculum. The kinetic parameters showed a specific growth rate (µ) of 0.11/d and doubling time (td) of 5.72 d, extending the stationary phase until day 24 (Figure 5). The percentage of viability of *P. carabiasiae* cell suspension remained at 100% from day 6 to day 24 of culture, with a reduction of 10% on day 30. This reduction coincides with the stage of cell death in growth kinetics.

### 2.3. Antioxidant and Chelating Activities, and Total Phenolic Content in Plant Extracts

In this work, the capacity of *P. carabiasiae* extracts to neutralize radicals via electron transfer (ABTS and FRAP) and to inhibit the production of hydroxyl radicals (chelating activity) were evaluated.

All extracts from *P. carabiasiae* cell suspension culture showed antioxidant activity by the FRAP method (Table 1), indicating that the extracts could protect the oxidative damage, inhibit hydrogen peroxide, and cause a marked reduction in the conversion of ferric to ferrous ions. However, no relevant activity was observed in the antiradical assay with ABTS. Among the three extracts, only the hexane extract showed chelating activity, but the percentage was 7.6 times lower than quercetin (*p* < 0.05). The results in Table 1 show that the antioxidant activity and total phenols content in the acetone extract were highest compared to the other extracts, suggesting that *P. carabiasiae* cells produce more molecules with antioxidant activity of an aprotic polar nature. This observation agrees with the results of the total phenolic content, where the acetone extract presented the highest value (30.57 ± 2.00 mg GAE/g DW).

### 2.4. Anti-Inflammatory Activity

The anti-inflammatory activity of the methanolic extract of *P. carabiseae* was concentration-dependent (Figure 6a). At the lowest concentration of the extract (10 μg/mL), the inhibition of nitrite (NO_2_^−^) production reached 73%, compared with LPS treatment alone, which yielded 118.77 ± 1.87 μM (maximum NO_2_^−^ production). The highest percentage inhibition of the extract and aminoguanidine was recorded at the concentration of 100 μg/mL (94 and 98%, respectively), with NO_2_^−^ values very close to the DMSO control (0.78 ± 0.11 μM). In all concentrations evaluated, the inhibition shown by aminoguanidine was significantly higher than that recorded by the same concentrations of *P. carabiseae* extract (*p* < 0.05).

The cell viability of macrophages treated with the different concentrations of *P. carabiseae* methanolic extract and aminoguanidine remained above 90% (Figure 6b), finding significant differences only with the 100 μg/mL concentration of the extract with respect to the DMSO control (*p* < 0.05).

## 3. Discussion

Micropropagation is a modern and reliable technology for the large-scale in vitro plant multiplication in a short time, since it allows obtaining high-quality plant material, protection of genetic resources, breeding, and stable production of bioactive secondary metabolites. In previous studies in the genus *Plukenetia*, in vitro germination, vegetative propagation, and callus induction have been reported in the *P. volubilis* species [15,16,17]. In this study, three *P. carabiseae* calluses lines were generated in B5 medium under different light regimes, obtaining yellow pigmented and friable calluses (Figure 2) with a maximum biomass at day 33 (Figure 3) in constant light conditions. This pigmentation could be due to the accumulation of carotenoids [18]. The intensity and light quality have a crucial role in secondary and primary metabolism, gene activity, and morphological processes in the in vitro culture system [19]. Similarly, Wahab et al. [17] obtained friable and greenly pigmented calluses of *P. volubilis* in MS medium with different combinations of 6-benzylaminopurine (BAP) and 2,4-dichlorophenoxyacetic acid (2,4-D), from ovule explants and 24 h light. The establishment of a cell suspension culture produced greater dry biomass and in half the cultivation time compared to callus culture under constant light (Figure 3 and Figure 5). This is the first report of growth kinetics characterization of a cell suspension culture in the genus *Plukenetia*. Some advantages of *P. carabiaseae* suspension culture to the secondary metabolites study are the scalability and the implementation of biotechnological strategies (addition of precursors or elicitors) to production enhancement [20].

Different polarity extracts from cell suspension registered low (hexane and aqueous) and high (acetone) antioxidant activity (Table 1). Compounds with antioxidant activity can prevent oxidative damage induced by free radicals, since oxidation of substrates occurs through a chain reaction involving three stages (initiation, propagation, and termination). Antioxidants arrest such stages through various mechanisms [21].

The hexane extract showed chelating activity (Table 1), which is important because it decreases the pro-oxidant effect of metals and can sterically prevent the formation of the metal hydroperoxide complex, protecting the cell [22]. Conversely, the high antiradical activity shown by the acetone extract coincides with the higher content of phenolic compounds, compared to the aqueous extract (Table 1). These results coincide with that reported by Puangpronpitag et al. [23], who analyzed extracts (aqueous, ethanolic, and hydroethanolic) of *P. volubilis* seedlings (21 day of growth), finding that the content of total phenols was higher in the ethanolic extract (23.08 mg/g extract) compared to the aqueous extract (1.5 mg/g extract), and that was related to a high antioxidant activity. The ethanolic extract showed the lowest IC_50_ values in the DPPH (0.007 mg/mL) and ABTS (1.4 mg/mL) assay. The FRAP activity of this extract was 2.6 times the activity recorded in the aqueous extract. On the other hand, Castaño et al. [8] reported that the constituents of *P. volubilis* leaves had the greatest contribution to antioxidant activity, showing that the lipophilic fraction of the ethanolic extract had the highest activity by the ABTS method (15.1 mg/100 mL extract). The preliminary chemical analysis of that ethanolic extract revealed the presence of terpenoids and phenolic compounds (flavonoids, anthraquinones, and coumarins), which could be involved in antioxidant activity. In the study of Nascimento et al. [3], the antioxidant activity of *P. volubilis* leaf extracts (250 μg/mL) was reported through the DPPH radical inhibition assay, which ranged from 62.8% (hexane extract) to 88.3% (aqueous extract), with the hydrophilic components being the most bioactive. On the other hand, total phenols, γ- and δ-tocopherols, α-linolenic, and linoleic acid have been identified as the most abundant components in seeds of sixteen *P. volubilis* cultivars, as well as good antioxidant capacity [9]. Therefore, antioxidant compounds of different natures have been reported in leaves and seeds of *Plukenetia* species. This supports the proposal from Wang et al. [24] that the antioxidant capacity of *P. volubilis* is affected by the processing method, origin, composition, and chemical properties of the biological material, considering that the optimization of the processing method could maximize the antioxidant potential of the extracts [25]. The total phenolic content in the acetone extract of *P. carabiasiae* is comparable to that shown in *Pistacia lentiscus* L. [26], considered one of the typical species of the Mediterranean undergrowth with the highest total phenolic content.

Additionally, the methanolic extract of the cell suspension culture showed a significant anti-inflammatory activity (Figure 6a) with NO_2_^−^ reduction percentages from 73 to 94% at the doses evaluated (10–100 µg/mL). Similar results were reported by More et al. [27], who recorded a reduction in NO_2_^−^ production of more than 80% with 100 μg/mL of *Elephantorrhiza elephantina* extract. The reduction of nitrite content can be carried out by direct neutralization of nitric oxide by extract components [28] and by inhibition of the expression or activity of the enzyme inducible nitric oxide synthase (iNOS) [29]. Other mechanisms of action of secondary metabolites with anti-inflammatory activity are the inhibition of secretion [30], enzyme activity [31], or the expression of genes [32] related to the synthesis of pro-inflammatory eicosanoids, as well as cyclooxygenase-2 (prostaglandins and thromboxanes) and lipooxygenase-15 (leukotrienes).

Secondary metabolites of different chemical nature with potential anti-inflammatory have been reported, as well as anthraquinones [31], flavonoids, stilbenes [30,33], lycopene [32], and alkaloids [34]. Although phytochemical information in the genus *Plukenetia* is limited, the presence of terpenoids, flavonoids, and coumarins has been reported [8]. Some antioxidant compounds act as inhibitors of the proteasome and nitric oxide production [30,35], which is related to its anti-inflammatory properties, so that phenolic compounds present in *P. carabiasiae* extracts (Table 1) may be responsible for their anti-inflammatory activity (Figure 6a). A bioguided study will be necessary for the isolation and identification of the compounds responsible for that activity.

The viability of murine macrophage cells was not affected by the addition of the methanolic extract, presenting a 92% at 100 μg/mL. Similarly, Flores-Vallejo et al. [36] reported the absence of cytotoxic effect of different fungal extracts on macrophage cells at the same concentration (100 μg/mL). This ensures the safe application of these extracts in a second stage of in vivo tests in murine models.

## 4. Materials and Methods

### 4.1. Plant Material

The fruits of *P. carabiasiae* were collected in the municipality of Yecuatla, Veracruz [5]. The seeds were separated and washed with a 3% solution of sodium hypochlorite (NaClO) for 8 min, and then with sterile distilled water three times.

### 4.2. Callus Generation

The seed coat was removed with a scalpel, and the seeds were sown in 10% Murashige & Skoog (MS) medium [37] with 3 g/L sucrose and 8 g/L agar, incubating at 25 °C. Subsequently, five leaf explants of the seedlings obtained in vitro were sown in the same MS medium added with the combination of growth regulators 2,4-dichlorophenoxyacetic acid (2,4-D, 3 mg/L), 6-benzylaminopurine (BAP, 2 mg/L), and 3-naphthaleneacetic acid (NAA, 2 mg/L), and incubated at 25 °C with a photoperiod of 16 h light/8 h dark, for 40 days [38].

### 4.3. Growth and Maintenance of Callus Culture

Callus propagation was performed in B5 medium supplemented with 30 g/L sucrose, 3 g/L polyvinylpyrrolidone, 1.5 g/L phytagel, and the combination of growth regulators 2,4-D (2 mg/L) plus kinetin (CIN, 2 mg/L). The pH of the medium was adjusted to 5.7 ± 0.2, and then the medium was sterilized at 121 °C for 15 min. The culture was incubated at 25 ± 2 °C and under three different lighting conditions: constant light (50 μmol/m^2^ s), photoperiod (16 h light/8 h darkness), and total darkness. All lines were sub-cultured every 20 days under the same conditions until friable calluses were obtained for more than 18 months.

For the growth kinetic analysis of the callus line, it was performed for 42 days, and samples were taken every 3 days. The biomass of each jar was freeze-dried, and the dry weight of the callus was recorded.

### 4.4. Establishment of Cell Suspension Culture

10 g of fresh weight of friable callus from the grown line were taken under constant light conditions and inoculated into 250 mL Erlenmeyer flasks with side baffles in Gamborg B-5 basal (B5) medium and with the combination of the 2,4-D: CIN growth regulators (2:2, mg/L). The culture was kept at 25 ± 2 °C in agitation (110 rpm) and constant light. Subcultures were performed every 2 weeks until fine cells were obtained, which were inoculated in Erlenmeyer flasks without baffles.

For growth kinetic analysis, the culture was maintained for 30 days, and three flasks were taken every 3 days to record the pH of the medium, cell viability, and dry weight (DW) of the biomass. For the dry weight, the content of each flask was filtered through medium-pore filter paper, the biomass was freeze-dried, and the dry weight of the cells was obtained. Cell viability was measured by the fluorescein diacetate (FDA) method with the support of a fluorescence microscope [39]. The specific growth rate (µ) was calculated from the slope of the straight line obtained by plotting ln x vs. time during the exponential growth phase and the doubling time (dt) by the equation dt = ln 2/µ).

### 4.5. Preparation of the Extracts

The freeze-dried biomass of the cell suspension (day 13 of culture) was pulverized and macerated for 15 min (Tissue Master 125-115-10; OMNI-INC, Kennesaw, GA, USA). Ultrasound-assisted extraction was performed using a tissue-solvent ratio 1:10 (*w*/*v*). To remove plant tissue, the extracts were centrifuged at 4000 rpm for 15 min at room temperature and the solvent was evaporated to give yields of 3.33 mg/g DW, 2.92 mg/g DW, 0.17 mg/g DW, and 0.18 mg/g DW in methanolic, hexane, acetone, and aqueous extracts, respectively. The hexane extract was subsequently re-suspended (2 mg/mL) in methanol for the following analyses.

### 4.6. Determination of Total Phenolic Compounds

The total phenolic content was analyzed using the Folin-Ciocalteu method [40]. The reaction mixture consisted of 1 mL of extract, 0.5 mL of Folin-Ciocalteu reagent, and after stirring, 0.4 mL of Na_2_CO_3_ (7.5%) was added. The mixture was stirred and incubated for 1 h at room temperature. The absorbance was read at 765 nm. The total phenolic content was expressed in mg gallic acid equivalent per g dry weight (mg GAE/g DW).

### 4.7. Antioxidant Activity

The inhibition of the radical 2,2′-azino-bis (3-ethylbenzothiazoline-6-sulfonic acid) (ABTS) was determined according to González-Palma et al. [41]. The ABTS radical was generated by the reaction of potassium persulfate (2.45 mM) and ABTS (7 mM) for 12–16 h at room temperature in the dark. The ABTS- radical solution was diluted with water to an absorbance of 0.70 at 734 nm. The reaction mixture comprised: 0.1 mL of extract and 2.9 mL of ABTS radical. It was incubated for 6 min, and the absorbance was read at 734 nm. Antioxidant activity was expressed as a percentage of inhibition calculated with Equation (1).% inhibition = [(Abs control − Abs sample)/Abs control] × 100(1)

The chelating activity of metal ions was performed according to the method previously described by Xie et al. [42]. One mL of extract was mixed with 50 μL of FeCl_2_ (2 mM), 1.85 mL of distilled water, and 100 μL of ferrozine (5 mM). It was kept at room temperature for 10 min and its absorbance was read at 562 nm. Chelating activity is reported as a percentage of inhibition, according to Equation (1).

The Ferric Reducing Antioxidant Power (FRAP) was performed using the procedure described by Sudha et al. [43]. The FRAP reagent was prepared with 2.5 mL of 10 mM TPTZ (2,4,6 tripyridyl-s-triazine) in 40 mM of HCl, 2.5 mL of FeCl_3_ 6H_2_O at 10 mM, and 25 mL of acetate buffer at 0.3 M (pH 3.6). The reaction mixture was performed with 900 μL of FRAP reagent, 80 μL of distilled water, and 20 μL of extract. The mixture was incubated at 37 °C for 10 min, then the absorbance was read at 593 nm. Results were expressed as µM trolox equivalent per g dry weight (μM trolox/g DW), according to a standard curve (5 to 20 μM/g).

### 4.8. In Vitro Anti-Inflammatory Assay

The anti-inflammatory property of the methanolic extract was evaluated by the murine macrophage cell line RAW 264.7 model, using nitrite (NO_2_^−^) detection as previously was described by León-Álvarez et al. [44]. RAW 264.7 is an adherent cell line isolated from a mouse tumor that was induced by Abelson murine leukemia virus. This cell line, with macrophage differentiation, can be used in oxidative stress, inflammatory, and antibacterial activity studies. RAW 264.7 is tumorigenic and is widely used in cancer and drug development research. This cell line was donated by the Biological Testing Laboratory of the Chemistry Institute of the National Autonomous University of Mexico (UNAM), which purchased it from the ATCC catalog.

RAW 264.7 cells (ATCC) were cultured in DMEM/Nutrient Mixture F-12 (DMEM/F12) medium supplemented with fetal bovine serum (10%) (GIBCO, Waltham, MA, USA) and incubated at 37 °C and 5% CO_2_. Cells were grown in 25 cm^2^ plates. Culture started with an initial inoculum of 10^6^ cells, and the culture medium was changed every two days.

After 7–8 days of starting the cell culture, cells were prepared in 96-well Falcon plates (100 µL/well, 2 × 10^5^ cell/L) to assess the anti-inflammatory activity of the methanolic extracts of lyophilized cell suspensions of *P. carabiasiae*, taking the point of greatest cell growth on day 13 of the kinetic. First, the cells were incubated at 37 °C and 5% CO_2_ for a period of 1 h. Subsequently, the different treatments were added in concentrations of 10, 50, and 100 µg/mL, which were incubated for 1 h. Afterwards, the cells were stimulated using lipopolysaccharide (LPS) from *Escherichia coli* serotype O111:B4 (Sigma-Aldrich, Saint Louis, MO, USA) at a concentration of 1 µg/mL, to induce an anti-inflammatory response. Finally, the cells were kept in incubation for a period of 24 h.

The production of NO_2_^−^ was used as a measure of inflammation due to the inhibition of inducible nitric oxide synthase (iNOS), usually activated during inflammatory processes. Aminoguanidine was selected as a positive control, as a direct inhibitor of iNOS and it was assayed at the same concentrations as the samples. Dimethyl sulfoxide (DMSO) was used as a negative control. After incubation for 24 h, 100 μL of the supernatant of each well was withdrawn, and 100 μL of Griess reagent was added (1% sulfanilamide and 0.1% naphthylenediamine dihydrochloride in 2.5% phosphoric acid). Plates were incubated for 10 min in the dark at room temperature, and absorbance was measured at 540 nm. The conditions of the cell viability assay in the cell lines were performed according to Flores-Vallejo et al. [36].

### 4.9. Statistical Analysis

Data on antioxidant activity (ABTS and FRAP), total phenolic content, and anti-inflammatory activity were analyzed using one-way analysis of variance (One-way ANOVA), and data transformation was used when necessary (log10). Pairwise mean comparisons were performed using Tukey’s test (*p* < 0.05). Student’s *t*-test was used to assess the statistical significance of differences in chelating activity data. Biomass and cell viability data from callus and cell cultures were reported as mean (*n* = 6) ± standard deviation of two independent experiments. All statistical analyses were performed in SigmaPlot software ( version 16.0).

## 5. Conclusions

In this study, we report for the first time the establishment of *Plukenetia carabiasiae* cell suspension culture from leaf explants, enabling the production of viable biomass and bioactive metabolites in shorter timeframes than callus culture. Also, it has potential as a constant and stable source of compounds with anti-inflammatory and antioxidant activity, the latter related to the content of phenolic compounds. Acetone extracts showed the highest phenolic content and antioxidant activity, while the methanolic extract exhibited potent inhibition of pro-inflammatory mediators without compromising macrophage viability. This is the first step in the phytochemical characterization of this species, which allows providing the basis for further studies focused on isolating and identifying specific metabolites, as well as for future applications in nutraceutical and pharmacological fields.

## Figures and Tables

**Figure 1 ijms-26-12190-f001:**
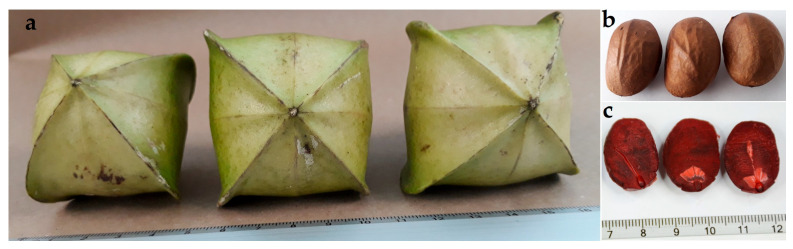
Fruits collected from *Plukenetia carabiasiae* (**a**), seeds with coat (**b**), and after scarification (**c**).

**Figure 2 ijms-26-12190-f002:**
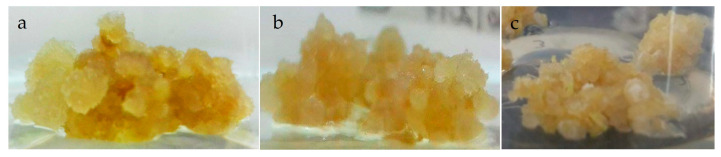
Cultures of *Plukenetia carabiasiae* calluses grown for 25 d in Gamborg basal (B5) medium (2,4-D: CIN) under conditions of (**a**) constant light, (**b**) photoperiod (16 h light/8 h darkness), and (**c**) darkness.

**Figure 3 ijms-26-12190-f003:**
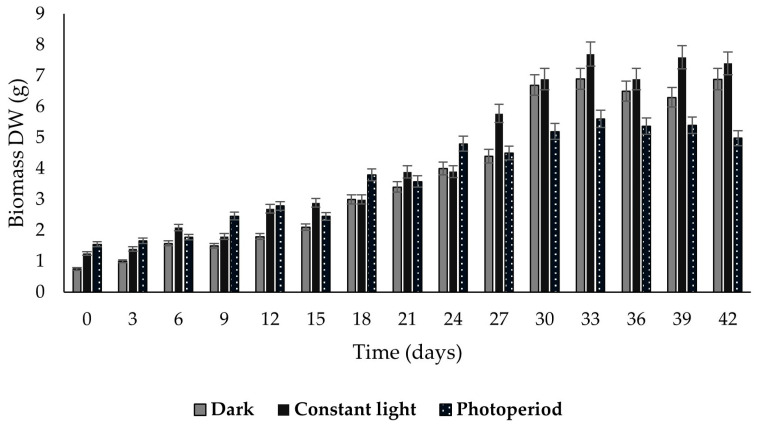
Growth kinetics of *Plukenetia carabiasiae* calluses in B5 medium (2,4-D: CIN) for 42 days under different light conditions. The data are expressed as mean values (*n* = 6) ± SD.

**Figure 4 ijms-26-12190-f004:**
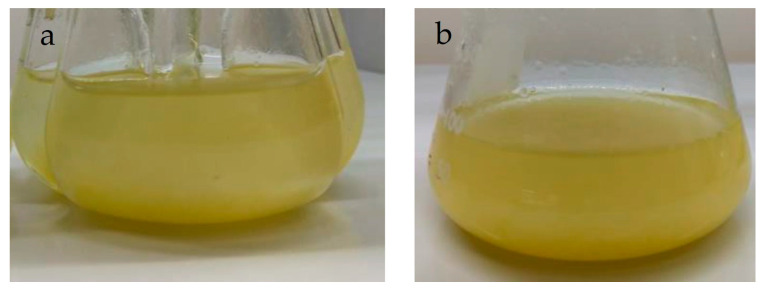
*Plukenetia carabiasiae* cell suspension cultures in B5 (2,4-D: CIN) medium, (**a**) in flasks with side baffles (establishment) and (**b**) after 30 days of growth.

**Figure 5 ijms-26-12190-f005:**
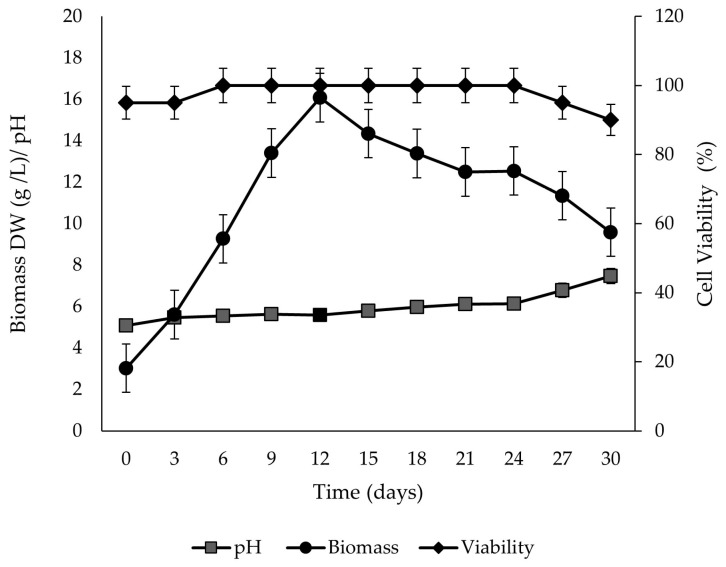
Growth kinetics, pH variation of the liquid medium, and viability (*n* = 6) of *Plukenetia carabiasiae* cell suspension culture in B5 (2,4-D:CIN) medium.

**Figure 6 ijms-26-12190-f006:**
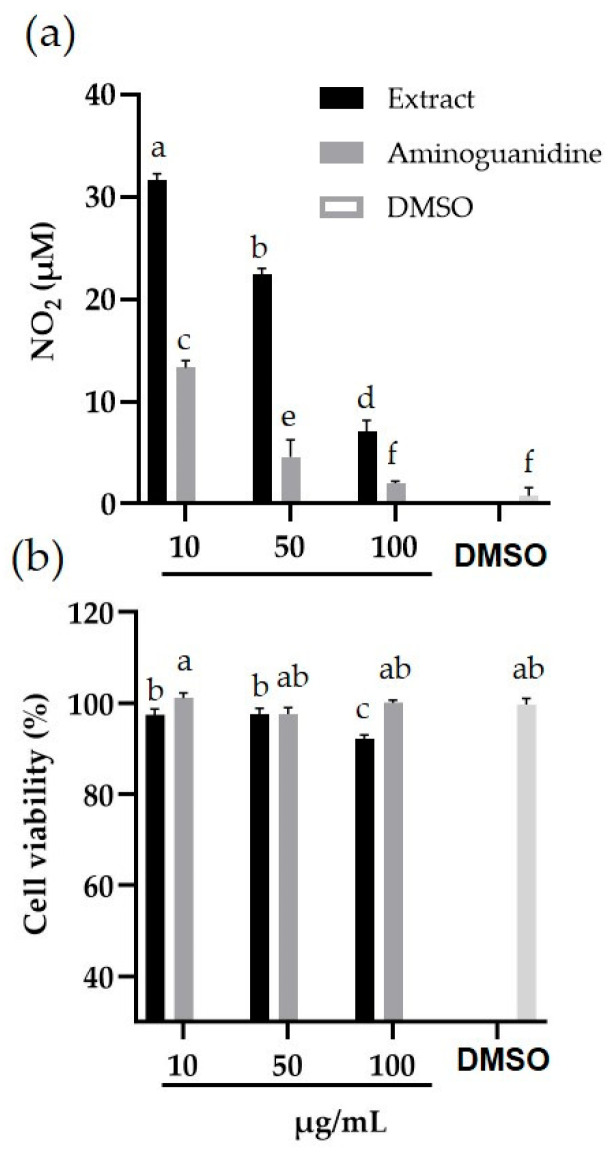
Inhibition of (**a**) nitric oxide (NO) production and (**b**) viability of the murine macrophage cell line RAW 264.7 induced by *E. coli* lipopolysaccharide, after treatment with aminoguanidine and *Plukenetia carabiasiae* methanolic extract (10, 50, and 100 μg/mL), or solvent dimethyl sulfoxide (DMSO). Mean values (*n* = 3–6) with different letters on each bar are statistically different (*p* < 0.05), according to Tukey’s test.

**Table 1 ijms-26-12190-t001:** Antioxidant activity and total phenols content in *Plukenetia carabiasiae* cell suspension extracts.

Assay	Aqueous Extract	Acetone Extract	Hexane Extract	Positive Control ^†^
ABTS (%)	4.37 ± 0.35 ^c^	16.97 ± 0.55 ^b^	0.38 ± 0.02 ^d^	70.00 ± 2.00 ^a^
Chelating activity (%)	0.00	0.00	5.14 ± 0.15 *	43.93 ± 1.29
FRAP (µM Trolox/g DW)	9.96 ± 0.13 ^b^	40.57 ± 0.86 ^a^	5.64 ± 0.35 ^c^	
Total phenols content (mg GAE/g DW)	9.42 ± 0.38 ^b^	30.57 ± 2.00 ^a^	7.84 ± 0.13 ^c^	

Values are means ± SD (*n* = 3). Mean values in the same row followed by different letters are statistically different, according to Tukey’s test (*p* < 0.05). ^†^ Quercetin (40 µg/mL). * Statistically different from the control at *p* < 0.05, according to Student’s test.

## Data Availability

The original contributions presented in this study are included in the article. Further inquiries can be directed to the corresponding author.

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
