# Peer review of "Antioxidant and Anti-Inflammatory Activity of Cell Suspension Culture Extracts of Plukenetia carabiasiae"

_ijms, 2025, doi:10.3390/ijms262412190_

Round 1
Reviewer 1 Report
Comments and Suggestions for Authors
- There are various species of the genus Plukenetia, why to use P. carabiasiae should be introduced.
- If possible, change Fig 2 with more professional photos.
- Please add the number of replicates for all the results.
- Have you tried alcohol as extract?
- The letters in Fig 5 b should be checked.
- The limit of current study should be discussed.
Author Response
- There are various species of the genus Plukenetia, why to use P. carabiasiae should be introduced.
-
R- P. carabiasiae is a species endemic to Mexico with a limited geographical scope and classified as endangered species and critically endangered, according to the calculation of area of occupancy (AOO) and extent of occurrence (EOO) (Villaseñor et al., 2024). The studies reported to P. carabiaseae are limited to the morphological description and distribution (Gutiérrez-Báez et al., 2020). This is the first study on the establishment of callus and cell suspension cultures, which is a necessary step to provide enough plant material for future bioprospection, as well as, phytochemical characterization of the extracts from the endemic species P. carabiasiae.
This information was added to the introduction.
Gutiérrez-Báez C., García-Morales S., Rodríguez-Domínguez J.M. and Zamora-Crescencio P. (2020). Plukenetia carabiasiae (Euphorbiaceae) nuevo registro para el estado de Veracruz, México. Desde el herbario CICY, 12, 144-148.
Villaseñor J.L., Ortíz E. and Murguía-Romero M. (2024). The rarest of rarities in the flora of Mexico. Botanical Sciences, 102(4), 1300-1317.
- If possible, change Fig 2 with more professional photos.
R- The resolution of Figure 2 has been modified.
- Please add the number of replicates for all the results.
R- The number of replicates has been added to the Materials and Methods section, and figures.
- Have you tried alcohol as extract?
R- Only the methanolic extract was evaluated for anti-inflammatory activity, while hexane, acetone, and aqueous extracts were evaluated for antioxidant activity. Ethanolic extracts have not been evaluated.
The letters in Fig 5 b should be checked.
R- The letters were revised and modified in the figure.
- The limit of current study should be discussed.
R- This study established the conditions for in vitro cultivation of this species endemic to Mexico. It also identified potential biological activities of pharmacological interest to promote its use and conservation. The perspective of this study is to analyze the extracts by HPLC-MS and its correlation with the antioxidant and anti-inflammatory activities.

Reviewer 2 Report
Comments and Suggestions for Authors
PC - Plukenetia carabiaseae
Abstract
"The aim of this work was to establish callus and cell suspension culture of P. carabiaseae, an endemic species of Mexico, ..."
=> It is expected that the reason is justified in the Intro part. Is PC hard to cultivate? What are the expectations for the callus/cell suspension culture? Is the callus /cell suspension culture economically justified for any antioxidant crop? What are the other reasons? Explain.
"After 35 d of cultivation, different polarity extracts from biomass were obtained, showing that the acetone extract had the highest antioxidant activity and high total phenolic content (30.57 mg of gallic acid equivalent (GAE) /g biomass)."
=> Was this experiment referenced to any ordinary PC extracts from standard-growth plants, or can it be referenced by comparison to the literature records?
=> How can we place this content among numerous "antioxidant" extracts?
=> Do you mean fresh or dry biomass? Supply the unit.
"This is the first study on the P. carabiaseae cell suspension culture and demonstrates its potential as a biotechnological source of antioxidant and anti-inflammatory metabolites."
=> Again: Is the callus /cell suspension culture economically justified for any antioxidant crop? Can we obtain, as humankind, such metabolites in any simpler way?
Intro
"The genus Plukenetia (Euphorbiaceae) comprises 25 species of lianas or vines distributed in the tropical regions of America, Africa, Asia, and Madagascar"
=> Add a suitable reference.
"These species are characterized by seeds of different sizes that are rich in omega-3 and -6 polyunsaturated fatty acids, as well as by their protein content (290 g kg-1)."
=> If protein content is estimated, do the same for fat.
"β-sitosterol is a major component in the ethanolic extract"
=> From the phytochemical point of view, it is hard to imagine. Depending on the analytical method, it can be the major component recognized then. Think it over.
"These oilseeds are consumed roasted, or their oil is used in the preparation of meals in the Amazon region"
=> Is it a kind of habit or rather an occasional practice?
=> What is the usual/expected crop yield of CP seeds per defined area?
"such as total phenols (64.6-80.0 mg GAE/100 g of seeds) and tocopherols (79-137 mg/100 g of seeds)."
=> Compare to results for common world crops to highlight the PC uniqueness.
==> Some issues/questions from the Abstract should be clarified/answered in this part.
Results
2.1
"The most friable calluses were observed in the line maintained in constant light and with yellow pigmentation (Figure 2a), followed by calluses grown in darkness (Figure 2c), highlighting their low pigmentation"
=> Is it a new observation or a standard? Discuss it with a suitable pro/contra/lack of reference.
2.3
"initiation (ABTS and reducing power) and propagation
=> Of what? End the sentence.
Table 1
Table 1 is uninterpretable/self-reproducible. We do not know the DER/DSR of the extracts. We do not know if the extracts are dried (like quercetin) or liquid (freshly prepared and tested). We do not know the concentrations of extracts and quercetin.
Discussion
"Different polarity extracts from cell suspension registered low (hexane and aqueous) and high (acetone) antioxidant activity (Table 1)."
+
"The hexane extract showed chelating activity"
=> How does it correspond to supposition on carotenoids presence in the callus? Carotenoids, powerful antioxidants, should be primarily present in the hexane extract.
Materials and Methods
"electric macerator"
=> Maceration, by definition, is a process that does not require energy consumption. Explain.
"using an sonication bath for 15 min, followed by 30 min of sonication"
=> I do not understand.
chapter 4.5
Using your description, it is impossible to calculate DER for any extract (or refer the mass of dried extract as the equivalent of dried callus mass in other way). We know that you have produced unknown amount of extract from unknown amount of plant material. And you analyze it in dried form.
=> It must be possible. Otherwise, all further experiments are uninterpretable.
chapter 4.6
Was the lipophilic hexane extract fully soluble in this assay to give you a reliable result?
chapter 4.7
Was the lipophilic hexane extract fully soluble in these assays to give you reliable results?
chapter 4.8
Why another type of extract was produced for this purpose? Its concentration/equivalency is again unknown.
Consider including simple HPLC(-MS/-DAD) characteristics of plant extract constituents. That is a standard practice required by most plant-oriented journals.
Comments on the Quality of English Languagesome exceptions (but look for other ones):
"Some reports in on the"
"anti-inflammatory activity recorded by the methanolic extract"
"three P. carabiseae calluses"
"an sonication bath"
"Alemania"
Author Response
Open Review 2
Comments and Suggestions for Authors
PC - Plukenetia carabiaseae
Abstract
"The aim of this work was to establish callus and cell suspension culture of P. carabiaseae, an endemic species of Mexico, ..."
=> It is expected that the reason is justified in the Intro part. Is PC hard to cultivate? What are the expectations for the callus/cell suspension culture? Is the callus /cell suspension culture economically justified for any antioxidant crop? What are the other reasons? Explain.
R- Plukenetia species are propagated by seeds, and the chances of genetic variations are high. P. volubilis shows poor seed viability, with a period from sowing to fruit ripening of nine months (Yang et al., 2014). The adult plants are susceptible to the root-knot nematode (Meloidogyne spp.) and fungal wild pathogens (Guerrero-Abad et al., 2021). All these factors influence the production of secondary metabolites, including antioxidant and anti-inflammatory metabolites. Several interesting pharmacological activities have been reported in P. volubilis, including antioxidant, antiproliferative, and anti-inflammatory properties. In contrast, other species, such as P. carabiasiae, remain unexplored.
- carabiasiae is a species endemic to Mexico with a limited geographical scope and is classified as an endangered species and critically endangered, according to the calculation of area of occupancy (AOO) and extent of occurrence (EOO) (Villaseñor et al., 2024). The studies reported to P. carabiaseae are limited to the morphological description and distribution (Gutiérrez-Báez et al., 2020).
The in vitro culture allows the stable production of bioactive compounds and the implementation of strategies to enhance their production as elicitation and scale-up in bioreactors (Fazili et al., 2022). However, all the information about the characterization of the oil seed and leaf extracts has been reported for the species P. volubilis. So, the establishment of callus and cell suspension cultures from the endemic species P. carabiasiae will allow to uncover its biological activities and the phytochemical characterization.
This information was added to the introduction.
Fazili M.A., Bashir I., Ahmad M., Yaqoob U. and Geelani S.N. (2022). In vitro strategies for the enhancement of secondary metabolite production in plants: a review. Bulletin of the National Research Centre, 46, 35.
Guerrero-Abad J.C., Padilla-Domínguez A., Torres-Flores E., López-Rodríguez C., Guerrero-Abad R.A., Coyne D., Oehl F., Corazon-Guivin M.A. (2021). A pathogen complex between the root knot nematode Meloidogyne incognita and Fusarium verticillioides results in extreme mortality of the inka nut (Plukenetia volubilis). Journal of Applied Botany and Food Quality, 94, 162–168.
Gutiérrez-Báez C., García-Morales S., Rodríguez-Domínguez J.M. and Zamora-Crescencio P. (2020). Plukenetia carabiasiae (Euphorbiaceae) nuevo registro para el estado de Veracruz, México. Desde el herbario CICY, 12, 144-148.
Villaseñor J.L., Ortíz E. and Murguía-Romero M. (2024). The rarest of rarities in the flora of Mexico. Botanical Sciences, 102(4), 1300-1317.
Yang C., Jiao D.Y., Geng Y.J., Cai C.T. and Cai Z.Q. (2014). Planting density and fertilisation independently affect seed and oil yields in Plukenetia volubilis L. plants. Journal of Horticultural Science & Biotechnology, 89(2), 201-207.
"After 35 d of cultivation, different polarity extracts from biomass were obtained, showing that the acetone extract had the highest antioxidant activity and high total phenolic content (30.57 mg of gallic acid equivalent (GAE) /g biomass)."
=> Was this experiment referenced to any ordinary PC extracts from standard-growth plants, or can it be referenced by comparison to the literature records?
R- In this study, an extract from wild plant material of P. carabiasiae was not obtained to compare with the results of the in vitro culture. However, in the discussion section, our results were compared with the antioxidant activity and phenolic content reported in extracts of seedlings (Puangpronpitag et al., 2021), leaves (Nascimento et al., 2013; Castaño et al., 2012), and seeds (Chirinos et al., 2013) from P. volubilis.
=> How can we place this content among numerous "antioxidant" extracts?
R- If we compare the total phenols content (TPC) recorded in the extracts of this study with those reported by Chaves et al. (2020), who analyzed extracts from 12 Mediterranean shrub species. The TPC content in the acetone extract of P. carabiasiae was similar to that shown in the species with the highest phenols content and antioxidant activity, including Pistacia lentiscus L., considered one of the typical species of the Mediterranean undergrowth with the highest total phenol content. This comparison was added to the discussion section.
Chaves N., Santiago A. and Alías J.C. (2020). Quantification of the antioxidant activity of plant extracts: Analysis of sensitivity and hierarchization based on the method used. Antioxidants, 9(76), 1-15. https://doi.org/10.3390/antiox9010076
=> Do you mean fresh or dry biomass? Supply the unit.
R- All results are based on dry weight of biomass, and it was specified throughout the document.
"This is the first study on the P. carabiaseae cell suspension culture and demonstrates its potential as a biotechnological source of antioxidant and anti-inflammatory metabolites."
=> Again: Is the callus /cell suspension culture economically justified for any antioxidant crop? Can we obtain, as humankind, such metabolites in any simpler way?
R- P. carabiasiae is classified as an endangered species, and an alternative source for obtaining its bioactive metabolites is through the in vitro culture. The results of this study demonstrated that P. carabiasiae cell suspension produces high levels of antioxidant and anti-inflammatory metabolites, which cannot be compared with any other study, due to the lack of phytochemical studies in this species. Nevertheless, in the discussion section, comparisons are made with the levels reported in P. volubilis and in other plant species. Based on this first study on P. carabiasiae, optimization studies can be carried out on extraction methods, types of solvents, elicitation strategies, etc.
Intro
"The genus Plukenetia (Euphorbiaceae) comprises 25 species of lianas or vines distributed in the tropical regions of America, Africa, Asia, and Madagascar"
=> Add a suitable reference.
R- The reference has been added in the introduction.
"These species are characterized by seeds of different sizes that are rich in omega-3 and -6 polyunsaturated fatty acids, as well as by their protein content (290 g kg-1)."
=> If protein content is estimated, do the same for fat.
R- The omega-3 and -6 contents have been added in the introduction.
"β-sitosterol is a major component in the ethanolic extract"
=> From the phytochemical point of view, it is hard to imagine. Depending on the analytical method, it can be the major component recognized then. Think it over.
R- The sentence has been modified and the β-sitosterol content has been added.
"These oilseeds are consumed roasted, or their oil is used in the preparation of meals in the Amazon region"
=> Is it a kind of habit or rather an occasional practice?
R- It is a traditional practice, the sentence has been modified for clarity.
=> What is the usual/expected crop yield of CP seeds per defined area?
R- Yang et al. (2014) reported seed yields from 1,549.2 to 2,154.6 kg/ha, depending on planting density, with a period from sowing to fruit ripening of nine months.
Yang C., Jiao D.Y., Geng Y. J., Cai C.T. and Cai Z.Q. (2014). Planting density and fertilisation independently affect seed and oil yields in Plukenetia volubilis L. plants. Journal of Horticultural Science & Biotechnology, 89(2), 201–207.
"such as total phenols (64.6-80.0 mg GAE/100 g of seeds) and tocopherols (79-137 mg/100 g of seeds)."
=> Compare to results for common world crops to highlight the PC uniqueness.
R- Total phenol content registered in five plant oils were from 2.84 to 14.44 mg GAE/g oil extract (Ahmed et al., 2024), whereas the analysis in sesame seed oil showed total phenol contents from 152.2 to 193.87 mg GAE/100 g seed oil (Oboulbiga et al., 2023). Taking into account that the organic oil content in sesame seeds ranges from 35 to 62% (w/w), the total phenols content reported in P. carabiasiae falls within the range reported for other oilseeds.
==> Some issues/questions from the Abstract should be clarified/answered in this part.
Results
2.1
"The most friable calluses were observed in the line maintained in constant light and with yellow pigmentation (Figure 2a), followed by calluses grown in darkness (Figure 2c), highlighting their low pigmentation"
=> Is it a new observation or a standard? Discuss it with a suitable pro/contra/lack of reference.
R- It is a new observation for the species P. carabiasiae, and the color and texture were parameters registered for the characterization of callus cultures. However, these results were similar in texture but not in color compared with the P. volubilis callus cultures. Wahab et al. (2023) reported the obtention of greenly pigmented calluses of P. volubilis under 24 h of light. The yellow pigmentation in P. carabiasiae calluses could be due to the accumulation of carotenoids (Gao et al., 2011), because the intensity and light quality have a crucial role in secondary and primary metabolism, gene activity, and morphological processes in the in vitro culture system (Cavallaro et al., 2022).
Cavallaro V., Pellegrino A., Muleo R., Forgione I. (2022). Light and plant growth regulators on in vitro proliferation. Plants, 11, 844-888.
Gao H., Xu J., Liu X., Liu B., Deng X. (2011). Light effect on carotenoids production and expression of carotenogenesis genes in citrus callus of four genotypes. Acta Physiol. Plant, 33, 2485–2492.
Wahab W.A., Jalila J.M., Rosle N., Rosli L.A. (2023). Meenakshi, T.C. In vitro callus induction of sacha inchi (Plukenetia volubilis), a PUFA-rich plant. RAS, 1(1445H), 88-100.
These comparisons with previous studies can be found in the discussion section.
2.3
"initiation (ABTS and reducing power) and propagation
=> Of what? End the sentence.
R- The sentence has been modified for clarity.
Table 1
Table 1 is uninterpretable/self-reproducible. We do not know the DER/DSR of the extracts. We do not know if the extracts are dried (like quercetin) or liquid (freshly prepared and tested). We do not know the concentrations of extracts and quercetin.
R- All extracts were obtained with a tissue:solvent ratio 1:10 (w/v), the extracts were centrifuged at 4000 rpm for 15 min, and the solvent was evaporated to give extraction yields of 3.33 mg/g DW, 2.92 mg/g DW, 0.17 mg/g DW and 0.18 mg/g DW in methanolic, hexane, acetone and aqueous extracts, respectively. The extracts were subsequently re-suspended in methanol (2 mg/mL) for the following analyses.
The concentration of quercetin used was 40 µg/mL. All this information was added to the Materials and Methods sections and the footnote of Table 1.
Results of total phenol content and the ferric reducing antioxidant power were expressed in mg per g dry weight.
Discussion
"Different polarity extracts from cell suspension registered low (hexane and aqueous) and high (acetone) antioxidant activity (Table 1)."
+
"The hexane extract showed chelating activity"
=> How does it correspond to supposition on carotenoids presence in the callus? Carotenoids, powerful antioxidants, should be primarily present in the hexane extract.
R- The presence of polyunsaturated fatty acids (linolenic, linoleic), tocopherols, and vitamin E have been reported in P. volubilis (Cardenas et al., 2021). Valencia et al. (2024) recorded a total carotenoid content from 0.6 to 1.5 mg/kg, while γ- and δ-tocopherol ranged between 861.6-1142 mg/kg and 587-717.1 mg/kg, respectively. Additionally, the total tocopherol content ranged from 1450 to 1856 mg/kg, confirming that carotenoids are not the main component. Pérez-Gálvez et al. (2020) have pointed out that carotenoids primarily act through radical scavenging (electron transfer reactions). Furthermore, the molecular structure of carotenoids (long hydrocarbon chains, with or without hydroxyl groups, in the case of xanthophylls) is not optimal to establish stable complexes with metals, compared to other antioxidants (flavonoids or citric acid), which have specific functional groups that allow them to bind strongly to metal ions. For this reason, we cannot claim that the chelating activity of the hexane extract from P. carabiasiae can be attributed to carotenoids.
Cárdenas, D. M., Gómez Rave, L. J., & Soto, J. A. (2021). Biological activity of sacha inchi (Plukenetia volubilis Linneo) and potential uses in human health: A review. Food Technology and Biotechnology, 59(3), 253-266.
Pérez-Gálvez, A., Viera, I., & Roca, M. (2020). Carotenoids and chlorophylls as antioxidants. Antioxidants, 9(6), 505.
Valencia, A., Muñoz, A. M., Ramos-Escudero, M., Chavez, K. C., & Ramos-Escudero, F. (2024). Carotenoid, tocopherol, and volatile aroma compounds in eight sacha inchi seed (Plukenetia volubilis L.) oil accessions. Journal of Oleo Science, 73(5), 665-674.
Materials and Methods
"electric macerator"
=> Maceration, by definition, is a process that does not require energy consumption. Explain.
"using an sonication bath for 15 min, followed by 30 min of sonication"
=> I do not understand.
R- The sentence has been modified for clarity.
chapter 4.5
Using your description, it is impossible to calculate DER for any extract (or refer the mass of dried extract as the equivalent of dried callus mass in other way). We know that you have produced unknown amount of extract from unknown amount of plant material. And you analyze it in dried form.
=> It must be possible. Otherwise, all further experiments are uninterpretable.
R- All extracts were obtained with a tissue:solvent ratio 1:10 (w/v), and the extraction yields were 3.33 mg/g DW, 2.92 mg/g DW, 0.17 mg/g DW, and 0.18 mg/g DW for the methanolic, hexane, acetone, and aqueous extracts, respectively
chapter 4.6
Was the lipophilic hexane extract fully soluble in this assay to give you a reliable result?
R- The extract was re-suspended in methanol at a concentration of 2 mg/mL, and it was sonicated until it was completely dissolved.
chapter 4.8
Why another type of extract was produced for this purpose? Its concentration/equivalency is again unknown.
R- Methanol is a solvent that allows us to perform an exhaustive extraction and evaluate the anti-inflammatory activity of this crude extract containing compounds of different chemical natures produced by the cell suspension. Subsequently, a bio-directed fractionation will be carried out.
Consider including simple HPLC(-MS/-DAD) characteristics of plant extract constituents. That is a standard practice required by most plant-oriented journals.
R- The present study established the conditions for in vitro cultivation of this species endemic to Mexico. It also identified potential biological activities of pharmacological interest to promote its use and conservation. The outlook is to analyze the extracts by HPLC-MS and its correlation with the antioxidant and anti-inflammatory activities, through collaboration with other institutions and research groups.
Comments on the Quality of English Language
some exceptions (but look for other ones):
"Some reports in on the"
"anti-inflammatory activity recorded by the methanolic extract"
"three P. carabiseae calluses"
"an sonication bath"
"Alemania"
R- The quality of the English language has been reviewed and the marked errors have been corrected.

Reviewer 3 Report
Comments and Suggestions for Authors
The authors investigated the antioxidant and anti-inflammatory activities of extracts prepared with various solvents from cell suspension cultures of Plukenetia carabiasiae.
The research topic is potentially interesting; however, the results presented are preliminary. I cannot recommend this manuscript for publication in its current form for the following reasons:
1) The study provides only limited novelty.
2) The results of the antioxidant assays are rather predictable, as some extracts obtained using different solvents are expected to exhibit measurable activity.
3) The phytochemical composition of the extracts was not analyzed; therefore, the authors were unable to establish correlations between bioactivity and specific compounds.
4) The cell suspension culture conditions were not optimized or modified to obtain more active extracts.
Given the preliminary nature of the data presented, a more comprehensive study is needed; for example, including an analysis of the metabolite composition of the extracts and the identification of the compounds responsible for the observed in vitro effects.
Conclusion: In its current form, the manuscript does not meet the publication standards of IJMS.
Author Response
Open Review 3
Comments and Suggestions for Authors
The authors investigated the antioxidant and anti-inflammatory activities of extracts prepared with various solvents from cell suspension cultures of Plukenetia carabiasiae.
The research topic is potentially interesting; however, the results presented are preliminary. I cannot recommend this manuscript for publication in its current form for the following reasons:
1) The study provides only limited novelty.
R- P. carabiasiae is classified as an endangered species (Villaseñor et al., 2024), and an alternative source for obtaining its bioactive metabolites is through the in vitro culture. The studies reported to P. carabiaseae are limited to the morphological description and distribution (Gutiérrez-Báez et al., 2020). The establishment of callus and cell suspension cultures are necessary steps to provide enough plant material for future bioprospection, as well as, the phytochemical characterization of the extracts.
2) The results of the antioxidant assays are rather predictable, as some extracts obtained using different solvents are expected to exhibit measurable activity.
R- If we compare the total phenols content (TPC) recorded in the extracts of this study with those reported by Chaves et al. (2020), who analyzed extracts from 12 Mediterranean shrub species, the TPC content in the acetone extract of P. carabiasiae was similar to that shown in the species with the highest phenols content and antioxidant activity, including Pistacia lentiscus L., considered one of the typical species of the Mediterranean undergrowth with the highest total phenol content.
Chaves N., Santiago A. and Alías J.C. (2020). Quantification of the antioxidant activity of plant extracts: Analysis of sensitivity and hierarchization based on the method used. Antioxidants, 9(76), 1-15. https://doi.org/10.3390/antiox9010076
3) The phytochemical composition of the extracts was not analyzed; therefore, the authors were unable to establish correlations between bioactivity and specific compounds.
R- A preliminary phytochemical composition of the extracts (total phenols content) was performed, and it was related to antioxidant activity. The perspective of this study is to analyze the extracts by HPLC-MS and its correlation with the antioxidant and anti-inflammatory activities.
4) The cell suspension culture conditions were not optimized or modified to obtain more active extracts.
R- The studies reported to P. carabiaseae are limited to the morphological description and distribution (Gutiérrez-Báez et al., 2020). The establishment of callus and cell suspension cultures will allow the phytochemical characterization of the extracts. Based on this first study on P. carabiasiae, optimization studies could be carried out on extraction methods, types of solvents, elicitation strategies, etc.
Gutiérrez-Báez C., García-Morales S., Rodríguez-Domínguez J.M. and Zamora-Crescencio P. (2020). Plukenetia carabiasiae (Euphorbiaceae) nuevo registro para el estado de Veracruz, México. Desde el herbario CICY, 12, 144-148.
Villaseñor J.L., Ortíz E. and Murguía-Romero M. (2024). The rarest of rarities in the flora of Mexico. Botanical Sciences, 102(4), 1300-1317.
Given the preliminary nature of the data presented, a more comprehensive study is needed; for example, including an analysis of the metabolite composition of the extracts and the identification of the compounds responsible for the observed in vitro effects.
Conclusion: In its current form, the manuscript does not meet the publication standards of IJMS.

Reviewer 4 Report
Comments and Suggestions for Authors
The use of plant cell cultures as potential sources of biologically active substances has attracted considerable interest among researchers for decades. The authors focused on this issue through a study of Plukenetia carabiasiae suspension cultures. Overall, the work is well-executed. Information is provided on their production and cultivation, assessment of cell growth and viability, determination of antioxidant activity, and the amount of polyphenols in three types of extracts (water, acetone, and hexane). The anti-inflammatory potential of its extracts of the extracts is also reported. The discussion reflects the purpose of the work.
However, the overall impression of this study is ambiguous, due to the following issues.
- Abstract. The text of the first two sentences (lines 20-23) is not relevant to the purpose of the work. If the study is devoted to assessing the activity of a suspension culture, then such a strong emphasis on callus production is unclear. If the authors consider this area of ​​research important, the title of the article should be changed.
- Introduction. Requires further revision and greater detail regarding in vitro cultures, their significance, and value.
- Results. Why and on what basis were the cultures grown in media with altered hormone compositions and gelling agents (agar, phytogel) during the study? This is not discussed. What was the criterion for these modifications?
- Why was a methanol used for anti-inflammatory potential of suspension extracts, which is not typical for other variants?
Author Response
Open Review 4
Comments and Suggestions for Authors
The use of plant cell cultures as potential sources of biologically active substances has attracted considerable interest among researchers for decades. The authors focused on this issue through a study of Plukenetia carabiasiae suspension cultures. Overall, the work is well-executed. Information is provided on their production and cultivation, assessment of cell growth and viability, determination of antioxidant activity, and the amount of polyphenols in three types of extracts (water, acetone, and hexane). The anti-inflammatory potential of its extracts of the extracts is also reported. The discussion reflects the purpose of the work.
However, the overall impression of this study is ambiguous, due to the following issues.
Abstract. The text of the first two sentences (lines 20-23) is not relevant to the purpose of the work. If the study is devoted to assessing the activity of a suspension culture, then such a strong emphasis on callus production is unclear. If the authors consider this area of ​​research important, the title of the article should be changed.
R- This is a newly established cell line for the species P. carabiasiae. The calluses were generated for the first time, and these were subsequently used to establish the cell suspension cultures.
Introduction. Requires further revision and greater detail regarding in vitro cultures, their significance, and value.
R- The information on in vitro culture in the introduction has been supplemented.
Results. Why and on what basis were the cultures grown in media with altered hormone compositions and gelling agents (agar, phytogel) during the study? This is not discussed. What was the criterion for these modifications?
R-The callus cultures were grown in the B5 medium, using half of the salts, which is more economical in a biotechnological process. Only two phytoregulators were used, which allowed the cells to grow with greater friability. The culture medium was supplemented with phytagel as was indicated in line 277.
Why was a methanol used for anti-inflammatory potential of suspension extracts, which is not typical for other variants?
R- Methanol is a solvent that allows us to perform an exhaustive extraction and evaluate the anti-inflammatory activity of this crude extract containing compounds of different chemical natures produced by the cell suspension. Subsequently, a bio-directed fractionation will be carried out.

Round 2
Reviewer 3 Report
Comments and Suggestions for Authors
The authors have adequately addressed the reviewers’ comments and revised the manuscript accordingly. Therefore, the manuscript can be accepted for publication in its current form.